# Gene Expression in Muscle-Invasive and Non-Muscle-Invasive Bladder Cancer Cells Exposed to Hypoxia

**DOI:** 10.3390/cancers17162624

**Published:** 2025-08-11

**Authors:** Rekaya Shabbir, Conrado G. Quiles, Brian Lane, Leo Zeef, Peter J. Hoskin, Ananya Choudhury, Catharine M. L. West, Tim A. D. Smith

**Affiliations:** 1Division of Cancer Sciences, University of Manchester, Manchester M20 4GJ, UK; rekaya.shabbir@manchester.ac.uk (R.S.); conrado.guerreroquiles@manchester.ac.uk (C.G.Q.); brian.lane@manchester.ac.uk (B.L.); peterhoskin@nhs.net (P.J.H.); ananya.choudhury@nhs.net (A.C.); catharine.west@manchester.ac.uk (C.M.L.W.); 2Bioinformatics Core Facility, University of Manchester, Manchester M13 6PT, UK; leo.zeef@manchester.ac.uk; 3Christie Hospital NHS Trust, Manchester M20 4BX, UK; 4Nuclear Futures Institute, School of Computer Sciences and Engineering, Bangor LL57 1UT, UK

**Keywords:** transcriptome, hypoxia, bladder cancer, signature biomarker

## Abstract

Bladder cancer can largely be divided into two types—non-muscle-invasive (NMIBC) and the more lethal form, muscle-invasive bladder cancer (MIBC). About 30% of NMIBC cases convert to MIBC. Predicting which patients with NMIBC are at risk of developing MIBC is crucial for determining the treatment course. Understanding the genetics of each of these cancer types enables the development of markers that can be used to predict conversion. In this study, we have measured genes that are upregulated by exposure to low oxygen levels (hypoxia)—a common characteristic of bladder cancers—to identify genes that may predict whether a bladder cancer is hypoxic and to identify potential markers of MIBC. We have found that different genes are increased in expression in NMIBC and MIBC, which may help to identify at-risk patients.

## 1. Introduction

Hypoxia is a characteristic of most solid cancers and confers an aggressive phenotype and treatment resistance, especially to radiotherapy [1]. Although radical cystectomy is considered the gold standard for treating patients with MIBC, some are treated with bladder-preserving radiotherapy in combination with a radiosensitiser [2,3]. Patients with hypoxic MIBC benefit from hypoxia modification [4]. Consequently, there is an interest in understanding the transcriptomic changes associated with hypoxia and developing biomarkers to stratify patients for treatment modification.

A bladder cancer-specific hypoxia-associated gene signature was derived by curating known hypoxia-sensitive genes, enrichment analysis in a bladder cancer cohort and final selection based on prognostication in a further clinical cohort [5]. The hypoxia score (HS) generated using the signature was validated as a prognostic marker in six independent bladder cancer cohorts [5] and was predictive of benefit from hypoxia modification in the BCON patient cohort. More recently, an in vitro study demonstrated the HS to be sensitive to hypoxia [6]. However, this study was only performed in two MIBC cell lines, and sensitivity to oxygen level was not determined.

Seed genes for hypoxia signatures can also be selected by identifying hypoxia-sensitive genes in cancer cell lines grown in vitro [7]. Previous work derived candidate gene panels by identifying genes upregulated by cells exposed to a single O_2_ level (1% O_2_) [8,9,10,11]. Tumours have a range of O_2_ levels from sub 1% O_2_. No one has investigated how the use of different oxygen levels might affect candidate gene panels and the ability to prognosticate. We hypothesised that identifying genes upregulated by exposing cells to a range of O_2_ levels would better represent the heterogeneity of oxygen levels in a tumour and provide a more representative gene panel for developing hypoxia-associated gene expression signatures. Therefore, we aimed to investigate how different oxygen levels affect transcriptional responses, derivation of gene signatures and signature prognostication.

In view of the propensity for NMIBC to progress to MIBC, gene candidate panels were derived based on cell lines derived from both NMIBC and MIBC [12]. As bladder cancer is a heterogeneous disease, reflected in its different molecular subtypes, multiple cell lines were used. Therefore, we generated gene expression data from four MIBC and two NMIBC cell lines exposed to different oxygen levels (20%, 1%, 0.2% and 0.1% O_2_) for 24 h.

## 2. Materials and Methods

### 2.1. Literature Review

A comprehensive systematic search for relevant studies and articles was conducted using keywords ((NMIBC OR MIBC OR muscle-invasive OR non-muscle-invasive) AND (bladder cancer)) AND (hypoxia AND gene expression) on PubMed. The search criteria, shown in the Appendix A, demonstrated that a comprehensive multiple bladder cancer cell line study of gene expression induced by cell exposure to 1%, 0.2% and 0.1% O_2_ levels had not previously been performed.

### 2.2. Cells

High-grade MIBC (J82, T24, UMUC3, HT1376) and low-grade NMIBC (RT4, RT112) bladder cancer cell lines obtained from the American Type Culture Collection were routinely screened for mycoplasma and authenticated. Characteristics of the cell lines are shown in Table 1. Cells were maintained in Eagle’s Minimum Essential Medium (EMEM) supplemented with 10% foetal bovine serum (FBS) (Gibco) and 2 mM L-glutamine in a humidified atmosphere of 95% air and 5% CO_2_ at 37 °C. EMEM was used for all cell lines because (1) it has a glucose content of 4.5mM, which is similar to the concentration found in human plasma, and (2) excess glucose affects the expression of glycolytic enzymes and other hypoxia-response genes [13].

### 2.3. Hypoxia Treatment

Cells (0.5 × 10^6^ cell/mL) were seeded with 10 mL of medium per Petri dish (area 56.7 cm^2^; Corning, Life Sciences, UK). Four Petri dishes per cell line and four oxygen conditions were used, except for the slow-growing RT112 cell line, where eight dishes per condition were used. Dishes were placed in an incubator at 95% air/5% CO_2_ at 37 °C for 24 h to allow for recovery from trypsin treatment. After 24 h, the medium was changed in the normoxic control dishes and returned to the incubator for 24 h. The remaining dishes were placed in the hypoxia cabinets at 1%, 0.2% and 0.1% O_2_ levels. Under hypoxic condition, the media in the dishes were washed twice with 5 mL of hypoxic medium (pre-incubated in the hypoxia cabinets [1%, 0.2% and 0.1% O_2_]). After adding 10 mL of the hypoxic medium to each dish, the dishes were incubated in hypoxia for 24 h. Each cell line/treatment experiment was caried out three times using different passages of cells (biological replicates).

### 2.4. Cell Harvesting

After 24 h of incubation, the cells were rinsed twice with PBS and harvested using a cell scraper (Corning^®^, Life Sciences, Loughborough, UK) by rotating and scraping in PBS. Ice-cold PBS hypoxia-equilibrated PBS was used. The harvested cells were transferred into 1 mL RNAse-free microfuge tubes (Eppendorf, Hamburg, Germany) and centrifuged at 4 °C for 10 min at ≥8000× *g*. Supernatants were removed and the cell pellets stored at −80 °C.

### 2.5. RNA Extraction

RNA was extracted using Qiagen QIA shredder Kit (Qiagen, Hilden, Germany) to break up the cells and the RNeasy Plus Mini Kit (Qiagen, Hilden, Germany) for RNA purification following the manufacturer’s protocol. RNA yield was determined using Nanodrop (Thermo-Scientific, Loughborough, UK) and Qubit following the manufacturers’ instructions. Mean and standard deviation of absorbance ratios 260/280 and 230/260, measures of RNA quality, for all RNA extracts were 2.07 ± 0.06 and 1.97 ± 0.12, respectively.

Clariom^TM^ S microarray gene expression arrays were prepared using 72 ng of RNA (8 ng/μL) with the Clariom S pico HT human assay. The samples were added to Kingfisher Shallow 96-well plates (Thermo Fisher Scientific, Loughborough, UK) along with brain and human prostate reference RNA (Thermo Fisher Scientific, Loughborough, UK), and RNA extracted from a clinical bladder cancer sample (plate control). YourGene Health (Manchester, UK) generated RNA data using the Clariom S pico HT human assay (Thermo Fisher Scientific, Loughborough, UK).

### 2.6. Data Normalisation and Batch Correction

The data generated from Clariom S underwent quality assessment (QA) and quality control (QC). The Affymetrix CEL files were normalised by single-space transformation with probe guidance cytosine count correction using Affymetrix Array Power Tools as recommended by the manufacturer (Thermo Fisher Sciences, Waltham, MA, USA) (https://assets.thermofisher.com/TFS-Assets/LSG/brochures/sst_gccn_whitepaper.pdf, accessed on 28 July 2025). Batch correction was performed by ComBat function on the R package ‘sva’. The full expression matrix was generated and log_2_-transformed.

### 2.7. Differentially Expressed Gene (DEG) Analysis

Microarray analysis was performed in R version 4.1.1 (10 August 2021). Gene expression values and normalisation were calculated using the RMA method [14]. Gene annotation was with affycoretools_1.64.0. and clariomshumantranscriptcluster. clariomshumantranscriptcluster.db: Affymetrix clariomshuman annotation data (chip clariomshumantranscriptcluster). R package (version 8.8.0.). Differential expression was calculated with limma_3.48.3 [15]. Adjusted *p*-values were corrected for multiple testing (Benjamini and Hochberg method).

### 2.8. Gene Expression Analyses

Two significance levels were used to select upregulated genes depending on the application. Biomarker development requires stringent inclusion criteria to reduce misdiagnosis. To derive biomarker candidate gene panels, we used *p_adj_* < 0.001 to select genes up- or downregulated by hypoxia to reduce the false positive rate (FDR). However, using tight criteria for identifying differentially upregulated genes between two disease subtypes may miss important genes, so a less stringent significance level (FDR), *p_adj_* < 0.05, was used to identify genes differentially regulated by hypoxia in MIBC and NMIBC cell lines.

In terms of the number of cells expressing each gene, to identify potential hypoxia biomarker candidates, genes upregulated in at least 3 cell lines were determined. To identify uniquely and commonly upregulated genes between MIBC and NMIBC cells, genes upregulated in both NMIBC cell lines were compared with genes upregulated in at least 2 MIBC cell lines.

The mean for each replicate was calculated and differences between cell lines investigated using unpaired *t* tests and GraphPad Prism 9 software Version 9.1.2. Biological processes were investigated using the PANTHER enrichment website. Heatmaps were generated with complexHeatmap v2.12.1 (PMID: 27207943). Gene ontology enrichment was studied using clusterProfiler (PMID: 34557778) and Enrichr v3.1 (PMID: 27141961). Gene Set Enrichment Analyses (GSEA) were performed using R version 4.1.1 (10 August 2021) ‘Limma package’.

### 2.9. Hypoxia Sensitivity in a Published Bladder Cancer Xenograft

The sensitivity of candidate hypoxia signature panels to hypoxia in an in vivo bladder cancer model was established using published gene expression data from a bladder cancer cell line-derived xenograft [16]. The data in this model was generated by sampling tissue from regions stained (hypoxic) and unstained (non-hypoxic) with the hypoxia probe pimonidazole injected into the host prior to tumour harvesting.

### 2.10. Hypoxia Scores

Expression and HSs were generated for each cell line using the 24-gene bladder hypoxia signature described above to verify hypoxia sensitivity. A candidate gene panel was derived by selecting genes upregulated by hypoxia in at least three cell lines. Hypoxia scores were generated using the median expression of the 24 genes comprising the West bladder cancer hypoxia signature [5]. To derive a cut-off value for patient stratification, the median value of hypoxia scores for the analysed patient cohort was determined, and tumours with higher or lower values were classed as hypoxic and normoxic, respectively.

### 2.11. Analysis of the Bladder Cancer TCGA Data Base and BCON

The complete dataset of the bladder TCGA cohort (*n* = 406) and the accompanying clinical data were downloaded from firebrowse.org. BCON data was previously generated [5]. Kaplan–Meier (KM) plots were used to analyse overall survival (OS) from the date of surgery to the date of death.

## 3. Results

Figure 1 shows a principal component analysis (PCA) of the cell line data (6 cell lines, 4 oxygen levels, 3 replicates per condition). This is based on differential gene expression at 55.7% (PC1) on *x*-axis separation of samples into normoxic on the right, with samples exposed to 1%, 0.2% and 0.1% O_2_ on the left broadly clustering with decreasing O_2_ concentration (although with some overlap). Cell line differences are apparent with 11.3% (PC1) with the NMIBC cell line RT4 at the top; next is RT112, with some co-clustering with HT1376 and J82, and then with UMUC3 and T24 clustering at the bottom of the *y*-axis. There was a separation between the NMIBC (RT4, RT112) and MIBC (HT1376, J82, T24, UMUC3) cell lines.

Figure 2 shows the expected trend for the upregulation of widely studied hypoxia-responsive genes (*CA9*, *SLC2A1* and *VEGF*) in the cell lines.

Table 2 summarises the number of genes affected by changes in oxygen levels, which increased with decreasing O_2_ level. More genes were upregulated than downregulated at all oxygen levels. Table 3 lists the genes up- or downregulated in at least three cell lines after exposure to 0.1% O_2_. (Appendix A show genes upregulated in two MIBC but no NMIBC cell lines and in both NMIBC but <2 MIBC cell lines by exposure to hypoxia, respectively). The number of processes affected by hypoxia increased with decreasing oxygen levels (Table 4). Cellular response to hypoxia was the only biological process affected by exposure to 1% O_2_. Additional processes affected by exposure to 0.2% O_2_ were carbohydrate transport and metabolism, including glycolysis, steroid hormone response and Peptidyl-proline hydroxylation to 4-hydroxy-L-proline, which is involved in collagen metabolism. More genes associated with glucose metabolism and with pregnancy were upregulated in cells exposed to 0.1% O_2_.

Genes also upregulated in 0.2% O_2_ are shown in bold and in 1% O_2_ in bold *.

Hypoxia scores were calculated using a 24-gene signature (HS_24_). HS_24_ tended to increase with decreasing oxygen levels (Figure 3). Figure 3B shows the median expression of the 77 genes (ME_77_) found to be upregulated in at least three bladder cancer cell lines after exposure to 0.1% O_2_ for 24 h. The median value of the 77 genes increased with decreasing oxygen levels in all cell lines. There were five genes in both signatures (*SYDE1*, *SLC2A3*, *DPYSL2*, *FUT11*, *P4HA2*). The median value of the five genes (ME_5_) also increased with decreasing oxygen level. As a further check of the hypoxia relevance of the generated signatures, we compared their expression in samples taken from an experimental tumour where hypoxia was identified using pimonidazole staining [15]. We showed previously that HS_24_ was higher in pimonidazole-positive areas. Appendix A shows ME_77_ and ME_5_ values were also higher. Appendix A shows which of the 24 genes were increased in expression by individual cell lines and O_2_ levels. Two genes, *CYP1B1* and *GLG1*, were not upregulated by hypoxia in any of the cell lines. Some cell lines differed only by one or two genes, but a core group of genes were upregulated by all cell lines.

Figure 4 shows a heatmap of the expression of the 78 (77 upregulated and 1 downregulated) genes found to be differentially expressed in at least three cell lines exposed to 0.1% O_2_ for 24 h among individual cell lines exposed to 1%, 0.2% and 0.1% O_2_ for 24 h. Most genes showed a gradual increase in expression with decreasing oxygen concentration (except *TAF8A*, which decreased).

Figure 5 shows Kaplan–Meier curves stratifying patients in the TCGA-BLCA cohort (*n* = 413) according to median HS_24_, ME_77_ and ME_5_. HS and ME scores were derived as the median gene expression of 24 genes (published signature), the 77 genes upregulated in ≥3 cell lines exposed to 0.1% O_2_ and the 5 genes common to both signatures, respectively. Both HS_24_ and ME_5_ performed similarly as prognostic biomarkers (*p* < 0.001) and were superior to ME_77_ (*p* = 0.056). Further Kaplan–Meier curves using several blader cancer cohorts are shown in Appendix A, demonstrating prognostication in some. However, ME_5_ did not predict a benefit of hypoxia modification (*p* = 0.77). ME scores derived from the 11 genes and 47 genes upregulated by exposure to 1% and 0.2% O_2_, respectively, were not prognostic for patients in the TCGA BLCA cohort (*p* = 0.5 and 0.32, respectively).

Biological processes associated with genes upregulated by at least two MIBC and both NMIBC cell lines or genes uniquely expressed by MIBC (at least two) or both NMIBC cell lines following exposure to hypoxia (0.1, 0.2 and 1% O_2_ for 24 h) were identified. Eight common genes were upregulated in ≥ 2 MIBC and in both NMIBC cell lines after exposure to 1% O_2_ (Figure 5). Additional genes were upregulated by exposure to 0.2% and 0.1% O_2_ (Table 5).

Table 6 shows the biological processes upregulated by exposure to each hypoxic O_2_ level A common to both NMIBC cell lines and at least two MIBC cell lines and (B) no NMIBC and at least two MIBC cell lines (exclusive to MIBC). Appendix A shows genes upregulated in at least 2 MIBC but not NMIBC cells and genes upregulated in both NMIBC cells but not in at least 2 MIBC cells exposed to (A) 0.1%, (B) 0.2% or (C) 1% O_2_ for 24. No biological processes were upregulated exclusively by NMIBC cells. Processes that were upregulated in ≥2 MIBC and both NMIBC cell lines by exposure to 1%, 0.2% and 0.1% O_2_ included response to hypoxia, protein hydroxylation, angiogenesis, VEGF signalling, processes related to glucose metabolism and amino acid metabolism, leucocyte chemotaxis and epithelial cell proliferation. Biological processes that were uniquely upregulated by MIBC cells were primarily pathways related to the negative control of kinases associated with intracellular signalling pathways.

## 4. Discussion

Cancer cells adapt to environmental stress such as hypoxia by upregulating the expression of genes encoding survival pathways. Identifying genes differentially regulated by hypoxia enables the derivation of hypoxia biomarkers that allows for the selection of patients for hypoxia modification. A hypoxia-sensitive panel of 77 genes commonly upregulated by at least three bladder cancer cell lines identified by exposing MIBC and NMIBC cell lines to 0.1% O_2_ compared unfavourably with the validated HS_24_ hypoxia biomarker. Five genes were identified, common to both the HS_77_ and HS_24_ gene panels, which proved to be equally as prognostic as the HS_24_ biomarker in the TCGA BLCA cohort but was not predictive for the selection of patients for hypoxia modification.

This study also determined biological pathways that were upregulated by exposure of NMIBC and MIBC cells to hypoxia. The biological process most significantly upregulated by both MIBC and NMIBC cells exposed to hypoxia was protein hydroxylation, driven by the expression of EGLN3, PLOD2, P4HA2, P4HA1 and EGLN1. Previous studies have demonstrated that the propyl-4-hydoxylases P4HA1 and 2 are upregulated by hypoxia in epithelial ovarian cancer cells exposed to hypoxia, and high expression is associated with decreased survival in patients with ovarian cancer. P4HA1 and 2 play a central role in collagen synthesis, EMT, collective cancer cell migration and angiogenesis [17,18,19]. PLOD2 encodes lysyl hydroxylase, which modifies collagen IV (COLVI), which modifies the extracellular matrix to weaken the lung endothelial barrier, is highly expressed in lung sarcoma metastasis [20]. PLOD2 is included in gene hypoxia signatures for RCC [21] and is a prognostic marker for patients with oesophageal cancer [22].

An important clinical requirement is the development of biomarkers that can identify NMIBC patients who are likely to progress to MIBC for early radical cystectomy or neoadjuvant/adjuvant chemotherapy to improve outcomes. Biological processes associated with genes upregulated uniquely by MIBC cells were predominantly negative regulation of kinase components of signalling pathways and driven by DUSP1, SPRY1, SPRY3, CAMK2N1 and PRKAR2B. The expression of the hypoxia-sensitive gene DUSP, which codes for dual specificity phosphatase, prevents HIF-1 overactivation by MAPK inhibition [23]. DUSP1 is prognostic for patients with gastric cancer [24] and important in immune cell regulation [23,24,25]. DUSP1 expression is decreased in glioblastoma-derived tumour stem cells (TSCs), but increased expression induces TSC differentiation [26]. PRKAR2B, which encodes Protein Kinase CAMP-Dependent Type II Regulatory Subunit Beta, is crucial for maintaining the growth of prostate cancer cells by activation of HIF-1α [27]. In lung cancer, calcium/calmodulin-dependent protein kinase II (CAMK2) inhibitor 1 (CAMK2N1) is a tumour suppressor gene which acts through inhibition of the Akt/mTOR pathway [28]. Overexpression of CAMK2N1 in lung cancer cells inhibits cancer cell proliferation and metastasis and increases cell death mechanisms [28]. However, in gastric cancer, CAMK2N1 expression has been shown to have a carcinogenic effect and to be negatively correlated with immune infiltrate [29], whilst in advanced prostate cancer (PCa) CAMK2N1 overexpression was associated with aggressiveness [30]. SPRY1 and 3 genes encode Sprouty 1 and 3, inhibitors of fibroblast growth factor (FGF). Expression of SPRY1 is associated with decreased growth and invasiveness of prostate and breast cancer [31,32]. However, SPRY3 is a designated tumour promoter for glioblastoma [33].

The expression of STC1 and LOX, which are only increased by hypoxia in MIBC cells, has previously been shown to have significant roles in human carcinogenesis [34,35,36]. STC1 expression is associated with poor postoperative survival [34], tumour growth and metastasis [35] and has been proposed as a potential drug target [34]. LOX expression has been shown to be associated with metastatic potential [36].

Expression of CA9 is commonly shown to be upregulated by hypoxia [37]. Whilst some studies have demonstrated association with prognosis [38,39], others have shown that the HIF-regulated genes *CA9* and HIF-specific prolyl hydroxylases (hypoxia-inducible PHDs) genes have limitations as hypoxia biomarkers [40,41,42,43]. The HIF-inducible gene VEGF was generally found to be increased in expression by hypoxia treatment, in agreement with previous work [44]. Several studies have demonstrated improvement in treatment response by inclusion of a VEGF inhibitor in several cancer types, including breast cancer [45], NSCLC [46], metastatic colorectal cancer [47] and renal cell carcinoma [48].

A metabolic switch in response to hypoxia indicates cancer progression [49]. The expression of glucose transporter members, including *SLC2A1* (GLUT1) and *SLC2A3* (GLUT3), was upregulated by hypoxia in most MIBC and NMIBC cells. Hypoxia-induced increase in GLUT1 protein expression has previously been shown to correspond with increased uptake of the PET glucose analogue FDG in hypoxic cancer cells [50]. The creatine transporter SLC6A8 was consistently upregulated in all cell lines. SLC6A8 promotes intracellular phosphocreatine levels in hypoxia, increasing ATP concentration, suppressing apoptosis and promoting survival [51]. An in vivo study has shown that inhibition of SLC6A8 inhibits colorectal cancer cell growth [52].

Based on a search in PubMed, no previous study has investigated the effects of exposure to 1%, 0.2% and 0.1% O_2_ concentration levels in multiple bladder cancer cell lines on gene expression. The three levels of O_2_ are associated with HIF1 induction (1% O_2_), UTR gene expression (0.2% O_2_) and of radiobiological relevance (0.1% O_2_). A possible criticism is that the study only examined cells exposed to hypoxia for a single time point, 24 h. However, by this time point, transcriptional adaptation to hypoxia has taken place, and 24 h represents biologically relevant hypoxia [53,54,55]. A fixed time point of 24 h is classed as chronic hypoxia, but some cells in solid cancers will be exposed to cycling hypoxia [56]. However, generally, cycling hypoxia induces similar but weaker changes in gene expression profiles than chronic hypoxia does [56]. The exception in some cells is the greater induction of immune-associated genes associated with cycling hypoxia [56].

This study examined changes in the expression of genes in established rather than primary bladder cancer cell lines in response to hypoxia. One study that examined gene expression in both established and primary bladder cancer cells lines demonstrated eight upregulated genes in primary bladder tissue of particular interest due to their sensitivity to hypoxia in vitro [57]. We found that seven of these genes, IGFBP3, VEGF, CCNG2, NDRG1, PFKFB3, ADM and SLC2A1, were also upregulated in MIBC and/or NMIBC in our study, suggesting parallel hypoxia-induced expression changes with primary bladder tumour tissue.

Two NMIBC cell lines were used in this study. There is one more NMIBC cell line available, SW780, which could be analysed in future work to verify the PCA clustering. Genomic analysis using in vivo models of NMIBC and MIBC cell lines would help verify differential gene expression.

## 5. Conclusions

In summary, this work has identified genes that are upregulated by exposure of multiple bladder cancer cells to hypoxia and demonstrated that some genes are upregulated in both MIBC and NMIBC cells but others uniquely in either MIBC or NMIBC cells. The study has also corroborated the hypoxia sensitivity of the HS_24_ bladder cancer gene signature in multiple cancer cell lines. The importance of gene expression platforms in biomarker design is also demonstrated.

## Figures and Tables

**Figure 1 cancers-17-02624-f001:**
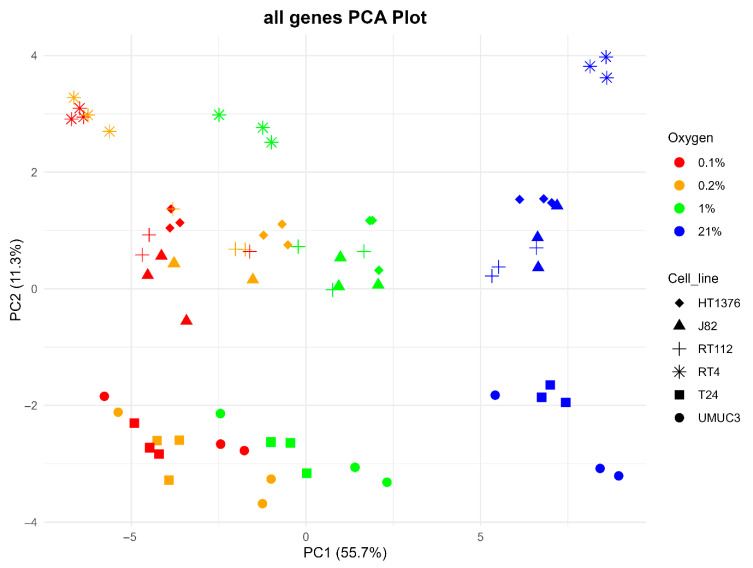
Principal component analysis of all genes expressed in each cell line/condition.

**Figure 2 cancers-17-02624-f002:**
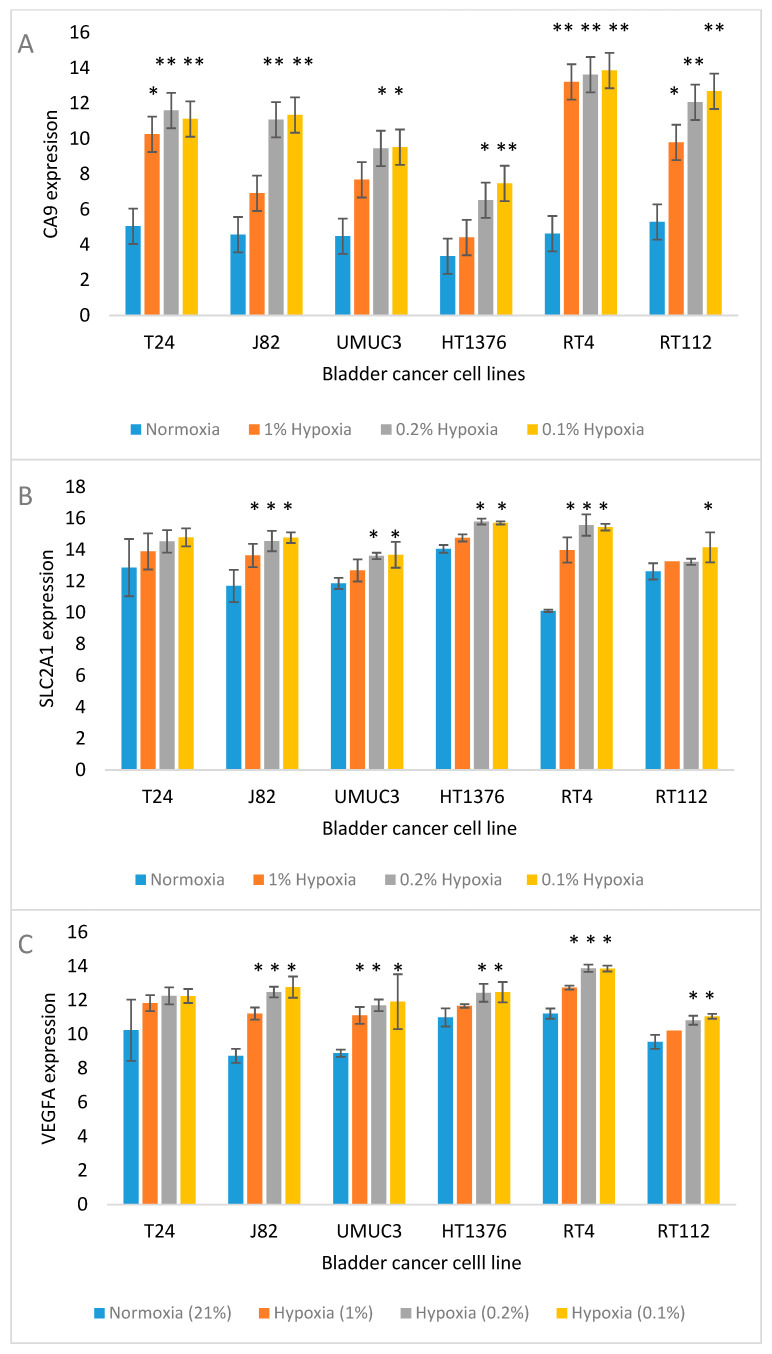
Expression of known hypoxia-associated genes, *CA9* (**A**), *SLC2A1* (**B**) and *VEGFA* (**C**) in bladder cancer cell lines incubated for 24 h in 21% (control), 1%, 0.2% and 0.1% O_2_. Significant differences from control * *p* < 0.05, ** *p* < 0.01.

**Figure 3 cancers-17-02624-f003:**
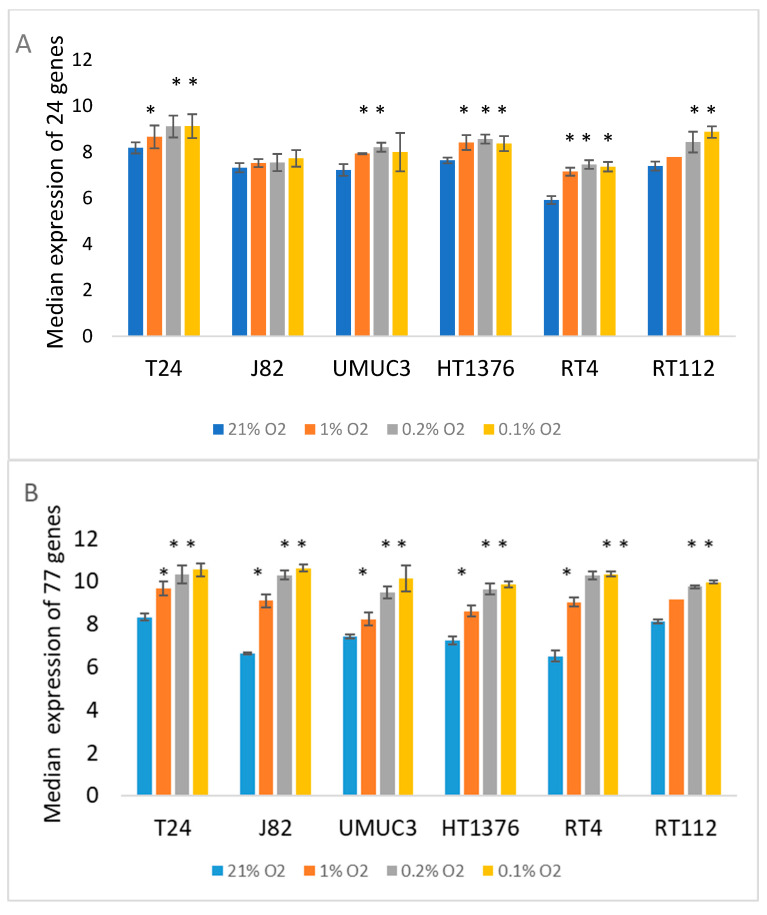
Sensitivity of gene expression-based hypoxia scores HS_24_ (**A**) and the median expression of 77 genes upregulated in at least 3 cell lines (HS_77_) (**B**) in bladder cancer cell lines incubated for 24 h in 21% (control), 1%, 0.2% and 0.1% O_2_. Significant differences from control * *p* < 0.05.

**Figure 4 cancers-17-02624-f004:**
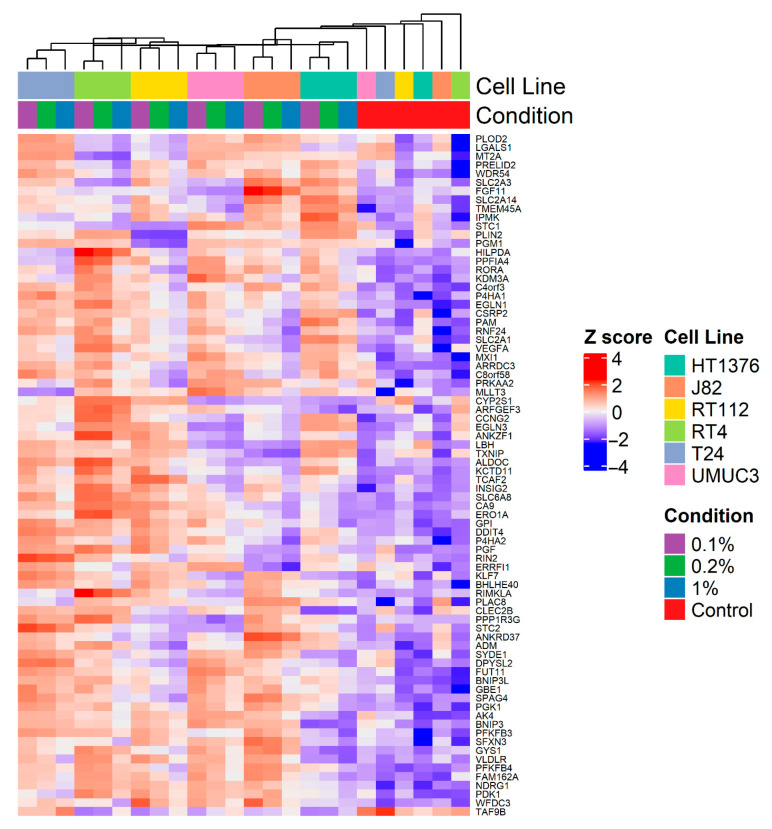
Heat map of expression of 78 genes (77 upregulated, 1 downregulated) in ≥3 bladder cancer cell lines exposed to 0.1% O_2_ for 24 h demonstrating gradation in expression in each cell line exposed to 1%, 0.2% and 0.1% O_2_ for 24 h.

**Figure 5 cancers-17-02624-f005:**
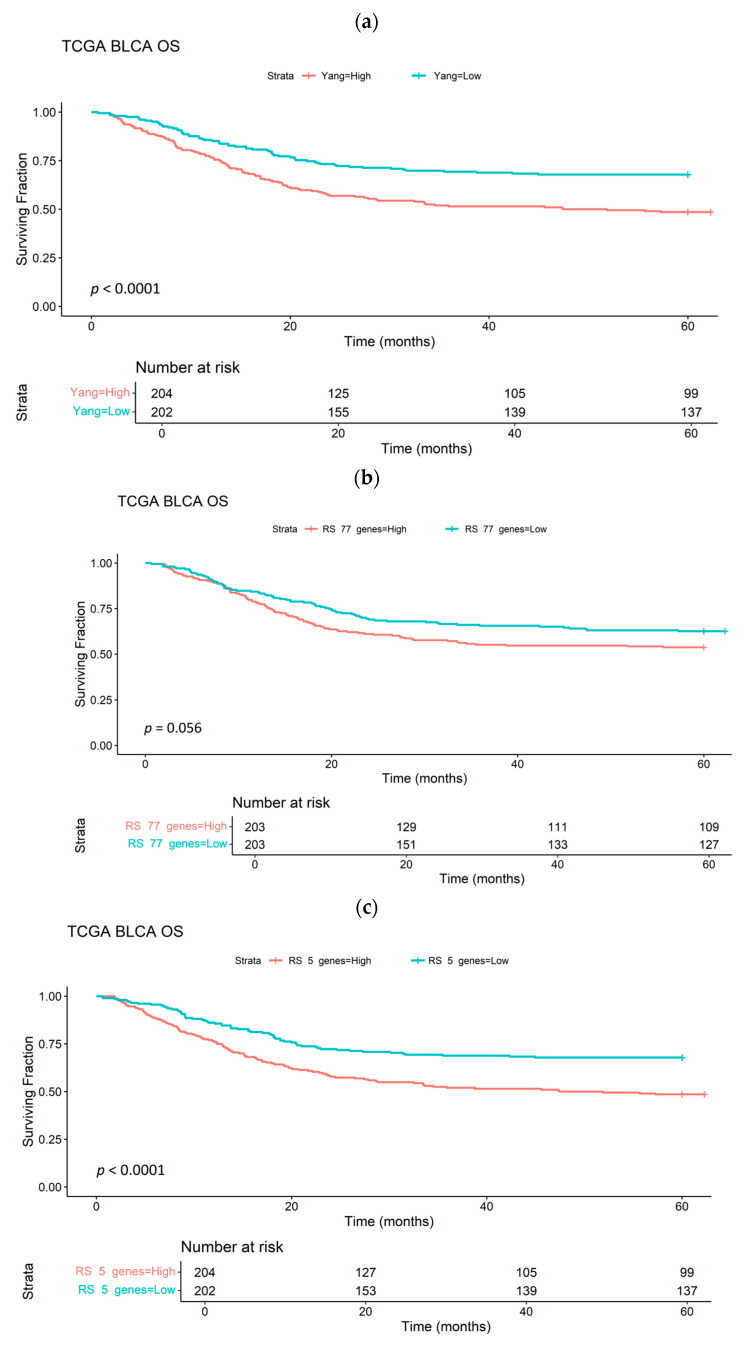
Kaplan–Meier survival curves for patients with bladder cancer in the TGCA-BLCA cohort (*n* = 408) stratified by median expression of a 24-gene signature (**a**), 77 genes upregulated in at least 3 bladder cancer cell lines exposed to 0.1% O_2_ (**b**) or 5 genes common to a and b (**c**).

**Table 1 cancers-17-02624-t001:** Characteristics of the NMIBC and MIBC cells used in this study.

Cell Line	Tissue Origin/Grade	Donor Age and Sex	Common Mutations	Overexpressed Receptors/Markers
**RT4**	Papilloma, non-invasive/grade I	63 M	Wild-type TP53/PTEN; FGFR3-driven luminal TERT and TSC1 promoter mutations; CDKN2A	FGFR; HER2; EN2 EGF; P2X1; P2X7; HRAS; PSCA
**RT112**	Non-muscle-invasive carcinoma grade 2		FGFR3 fusion/amplification; PIK3CA; wild-type PTEN; SRC; TERT promoter mutation	FGFR; EN2 CKs; FGFR3; PSCA
**HT-1376**	Invasive transitional cell carcinoma/ grade 3	58 F	Loss of CDKN2A/2B (9p21), likely TP53 alteration, typical invasive type; HRAS	EGFR; CD46; GPR87; CD74
**T24**	Muscle-invasive carcinoma/grade 3	81 F	TP53 mutant; PTEN mutant; TERT promoter mutation; HRAS	GRβ; EN2; ZEB1/2; EGFR; TRPM7
**UMUC-3**	Muscle-invasive carcinoma/grade 3	M	TP53 mutant; PTEN mutant + partial deletion; ATM mutation; KRAS	EN2; GRβ; ZEB1/2; TPβ
**J82**	Muscle-invasive carcinoma/grade 3	58 M	TP53 mutant; PTEN mutant + partial deletion; PI3KCA	EN2; ZEB1/2; EGFR; RON

CD, cluster of differentiation; EGF, epidermal growth factor; EN2, engrailed-2 transcription factor; FGFR, fibroblast growth factor receptor; GPR, G-protein coupled receptor; GRβ, glucocorticoid receptor; HER2, human epithelial growth factors; Hras, H-Harvey rat sarcoma virus; P2X, purinergic receptors; PSCA, prostate stem cell antigen; TPβ, Thromboxane receptor; TRPM7, transient receptor potential melastatin; ZEB1/2, zinc-finger E-box-binding homeobox (extracellular matrix markers).

**Table 2 cancers-17-02624-t002:** Number of genes affected by changes in oxygen levels in 1–6 of the cell lines: RT4, RT112, HT1367, T24, J82 and UMUC3.

Oxygen Level vs. Control	Number of Gene Affected in 1–6 Cell Lines					
	1	2	3	4	5	6
Up- and downregulated						
1%	239	42	11	4	0	0
0.2%	1014	145	47	20	8	2
0.1%	1264	216	78	33	18	3
Upregulated						
1%	178	42	11	4	0	0
0.2%	589	131	47	20	8	2
0.1%	739	178	77	33	18	3
Downregulated						
1%	61	0	0	0	0	0
0.2%	425	14	0	0	0	0
0.1%	525	38	1	0	0	0

**Table 3 cancers-17-02624-t003:** Genes up- or downregulated in at least three cell lines exposed to 0.1% O_2_. Bold upregulated by 0.2% O_2_; bold and * upregulated by 1% O_2_.

*Gene*						
** *ADM* **	** *CA9 ** **	*FGF11*	*LGALS1*	** *PGK1* **	** *RORA* **	** *TMEM45A ** **
** *AK4 ** **	** *CCNG2* **	** *FUT11* **	** *MLLT3* **	*PGM1*	** *SFXN3* **	** *TXNIP* **
** *ALDOC* **	*CLEC2B*	*GBE1*	*MT2A*	** *PLAC8* **	** *SLC2A1* **	*VEGFA*
** *ANKRD37* **	** *CSRP2* **	** *GPI* **	** *MXI1 ** **	** *PLIN2* **	** *SLC2A14* **	*VLDLR*
** *ANKZF1* **	*CYP2S1*	*GYS1*	** *NDRG1 ** **	** *PLOD2* **	** *SLC2A3* **	*WDR54*
*ARFGEF3*	** *DDIT4 ** **	*HILPDA*	** *P4HA1* **	*PPFIA4*	** *SLC6A8* **	** *WFDC3* **
** *ARRDC3* **	*DPYSL2*	** *INSIG2* **	** *P4HA2* **	*PPP1R3G*	** *SPAG4* **	
*BHLHE40*	** *EGLN1 ** **	*IPMK*	*PAM*	*PRELID2*	** *STC1* **	
** *BNIP3 ** **	*EGLN3*	** *KCTD11* **	*PDK1*	*PRKAA2*	*STC2*	
** *BNIP3L* **	*ERO1A*	** *KDM3A* **	** *PFKFB3 ** **	*RIMKLA*	** *SYDE1* **	
** *C4orf3* **	** *ERRFI1* **	*KLF7*	** *PFKFB4 ** **	** *RIN2* **	*TAF9B*	
*C8orf58*	** *FAM162A* **	** *LBH* **	*PGF*	*RNF24*	** *TCAF2 ** **	

**Table 4 cancers-17-02624-t004:** Biological processes associated with genes upregulated by at least three bladder cancer cell lines exposed to 0.2% and 0.1% hypoxia for 24 h.

O_2_ Level	PANTHER GO-Slim Biological Process	Fold Enrichment	FDR
1%	Cellular response to hypoxia	58	1.8 × 10^−3^

0.2%	Protein hydroxylation (GO: 0018126)	>100	3.9 × 10^−5^
0.2%	Glucose transmembrane transport (GO: 1904659)	>100	0.0027
0.2%	Response to hypoxia (GO: 0001666)	71.46	0.005
0.2%	Hexose biosynthetic process (GO: 0019319)	66	0.023
0.2%	Vitamin transport (GO: 0051180)	64.3	0.004
0.2%	Glycolytic process (GO: 0006096)	55.9	0.0048
0.2%	Apoptotic mitochondrial changes (GO: 0008637)	45.1	0.04
0.2%	Mitochondrial membrane organization (GO: 0007006)	45.1	0.039
0.2%	Mitochondrial transport (GO: 0006839)	19.5	0.027
0.2%	L-amino acid metabolic process (GO: 0170033)	18.9	0.029
0.2%	Cellular modified amino acid metabolic process (GO: 0006575)	16.3	0.039
0.2%	Organic anion transport (GO: 0015711)	10.9	0.01
0.1%	Protein hydroxylation (GO: 0018126)	98.9	7 × 10^−7^
0.1%	Response to hypoxia (GO: 0001666)	85.8	1 × 10^−7^
0.1%	Sprouting angiogenesis (GO: 0002040)	73.5	0.026
0.1%	Positive regulation of epithelial cell proliferation (GO: 0050679)	64.3	0.03
0.1%	Vascular endothelial growth factor receptor signaling pathway (GO: 0048010)	64.3	0.03
0.1%	Glucose transmembrane transport (GO: 1904659)	64.31	0.007
0.1%	Positive regulation of leukocyte chemotaxis (GO: 0002690)	46.8	0.039
0.1%	Hexose biosynthetic process (GO: 0019319)	39.6	0.043
0.1%	Vitamin transport (GO: 0051180)	38.6	0.01
0.1%	Glycolytic process (GO: 0006096)	33.6	0.012
0.1%	L-amino acid metabolic process (GO: 0170033)	15.1	0.017
0.1%	Cellular modified amino acid metabolic process (GO: 0006575)	13.0	0.026
0.1%	Apoptotic process (GO: 0006915)	12.4	0.027
0.1%	Regulation of multicellular organismal development (GO: 2000026)	9.03	0.04
0.1%	Organic anion transport	6.5	0.04
0.1%	Peptidyl-amino acid modification (GO: 0018193)	5.72	0.03

**Table 5 cancers-17-02624-t005:** Genes that are upregulated in at least two MIBC and both NMIBC cell lines exposed to 1%, 0.2% and 0.1% O_2_ for 24 h compared with cells maintained in 21% O_2_.

Upregulated at 1%						
*AK4*	*BNIP3*	*DDIT4*	*EGLN3*	*NDRG1*	*PFKFB4*	*TMEM45A*
	*WFDC3*					
**Upregulated at 0.2%**						
*ADM*	*BNIP3L*	*DPYSL2*	*KCTD11*	*PAM*	*PLAC8*	*SLC2A14*
*AK4*	*C4orf3*	*EGLN3*	*KDM3A*	** *PDK1* **	*PLOD2*	*SLC2A3*
*ALDOC*	*CA9*	*FAM162A*	*MXI1*	*PFKFB3*	** *PPFIA4* **	*SLC6A8*
*ANKZF1*	*CCNG2*	*FUT11*	*NDRG1*	*PFKFB4*	** *RIMKLA* **	*SPAG4*
** *ARID3A* **	*CSRP2*	*GPI*	*P4HA1*	*PGF*	*RIN2*	*SYDE1*
*BNIP3*	*DDIT4*	*INSIG2*	*P4HA2*	*PGK1*	*SFXN3*	*TCAF2*
*VLDLR*	*WFDC3*					
**Upregulated at 0.1%**						
*ADM*	*BNIP3L*	*EGLN3*	*INSIG2*	*P4HA2*	*PNRC1*	*SNX33*
*AK4*	*C4orf3*	*ERO1A*	*KCTD11*	*PAM*	*RIN2*	*SPAG4*
*ALDOC*	*CA9*	*FAM162A*	*KDM3A*	*PFKFB3*	*RNF24*	*SYDE1*
*ANKRD37*	*CCNG2*	*FOSL2*	*LRP1*	*PFKFB4*	*RORA*	*TCAF2*
*ANKZF1*	*CSRP2*	*FUT11*	*MRPL23*	*PGF*	*SFXN3*	*TMEM45A*
*ARRDC3*	*DDIT4*	*GBE1*	*MXI1*	*PGK1*	*SLC2A14*	*VLDLR*
*BHLHE40*	*DPYSL2*	*GPI*	*NDRG1*	*PLAC8*	*SLC2A3*	*WFDC3*
*BNIP3*	*EGLN1*	*HILPDA*	*P4HA1*	*PLOD2*	*SLC6A8*	*YEATS2*

**Table 6 cancers-17-02624-t006:** Biological pathways associated with genes upregulated by exposure to 1%, 0.2% and 0.1% O_2_ for 24 h in (**A**) at least 2 MIBC and both NMIBC cell lines and (**B**) at least 2 MIBC but no NMIBC cell lines.

	(A).		
O_2_ Level	PANTHER GO-Slim Biological Process	Fold Enrichment	FDR
1%	Response to hypoxia (GO: 0001666)	>100	0.043

0.2%	Protein hydroxylation (GO: 0018126)	>100	0.000033
0.2%	Response to hypoxia (GO: 0001666)	99.42	0.00007
0.2%	Glucose transmembrane transport (GO: 1904659)	74.57	0.02
0.2%	Hexose biosynthetic process (GO: 0019319)	68.83	0.02
0.2%	Glycolytic process (GO: 0006096)	58.4	0.006
0.2%	Apoptotic mitochondrial changes (GO: 0008637)	47	0.03
0.2%	Mitochondrial membrane organization (GO: 0007006)	47	0.03
0.2%	Vitamin transport (GO: 0051180)	44.7	0.04
0.2%	mitochondrial transport (GO: 0006839)	20.3	0.021
0.2%	L-amino acid metabolic process (GO: 0170033)	19.74	0.022
0.2%	Cellular modified amino acid metabolic process (GO: 0006575)	17	0.03
0.2%	Organic anion transport (GO: 0015711)	9	0.035
0.2%	Peptidyl-amino acid modification (GO: 0018193)	8.29	0.02

0.1%	Protein hydroxylation (GO: 0018126)	>100	3.7 × 10^−7^
0.1%	Response to hypoxia (GO: 0001666)	>100	1.2 × 10^−6^
0.1%	Glucose transmembrane transport (GO: 1904659)	60.2	0.026
0.1%	Hexose biosynthetic process (GO: 0019319)	55.55	0.028
0.1%	Glycolytic process (GO: 0006096)	47.09	0.01
0.1%	Apoptotic mitochondrial changes (GO: 0008637)	38	0.047
0.1%	Mitochondrial membrane organization (GO: 0007006)	38.01	0.047
0.1%	L-amino acid metabolic process (GO: 0170033)	21.24	0.009
0.1%	Cellular modified amino acid metabolic process (GO: 0006575)	18.28	0.01
0.1%	Mitochondrial transport (GO: 0006839)	16.41	0.037
0.1%	Peptidyl-amino acid modification (GO: 0018193)	6.69	0.038

	**(B).**		
**O_2_ Level**	**PANTHER GO-Slim Biological Process**	**Fold Enrichment**	**FDR**
0.2%	Carbohydrate metabolic process (GO:0005975)	13.65	0.011

0.1%	Negative regulation of protein kinase activity (GO:0006469)	28.50	0.0009
0.1%	Glycolytic process (GO:0006096)	28.3	0.018
0.1%	Negative regulation of MAPK cascade (GO:0043409)	23.21	0.03
0.1%	Regulation of protein serine/threonine kinase activity (GO:0071900)	14.94	0.02

## Data Availability

Cell line transcriptomic data generated in this study will be made publicly available on the GEO repository (accession number GSE302555).

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
