# Peer review of "Gene Expression in Muscle-Invasive and Non-Muscle-Invasive Bladder Cancer Cells Exposed to Hypoxia"

_cancers, 2025, doi:10.3390/cancers17162624_

Round 1
Reviewer 1 Report
Comments and Suggestions for Authors
This manuscript describes analysis of gene expression under the hypoxia in several bladder cancer cell lines (NMBC and MIBC). The clear aim is to define genes that can be a predictive marker in BC progression. It is very important and contemporary task. The approach combines data analysis from databases and literature as well as from this study. The limitation is using in experiments cell lines only. However, this study could be considered as an additional step to better understanding of mechanisms of BC progressions.
Author Response
Dear Reviewer,
Thank you for your helpful comment.
We have now included a paragraph describing a study that looked at the expression of hypoxia genes in primary bladder tumours. We noted that the expression of 7 of the 8 genes they identified as upregulated in their >5 of the 32 primary tumours they examined were upregulated in at least 3 of our cell lines or specifically in the MIBC cell lines.
Reviewer 2 Report
Comments and Suggestions for Authors
Dear authors,
thank you for providing your paper "Gene Expression in Muscle Invasive and Non-muscle Invasive Bladder Cancer Cells Exposed to Hypoxia".
The topic appears to be relevant in a clinical context, even though it concerns basic research. You can find my comments and suggestions for improvement below.
Introduction:
"About half of patients with muscle invasive bladder cancer (MIBC) are treated with radiotherapy in combination with a radiosensitiser [2,3]. "
That is not a generally true number, e.g. In Germany 75% of patients undergo radical cystectomy (Flegar L, Kraywinkel K, Zacharis A, Aksoy C, Koch R, Eisenmenger N, Groeben C, Huber J. Treatment trends for muscle-invasive bladder cancer in Germany from 2006 to 2019. World J Urol. 2022 Jul;40(7):1715-1721. doi: 10.1007/s00345-022-04017-z. Epub 2022 Apr 29. PMID: 35486177; PMCID: PMC9237006.)
Please specify/check your citations or change the statement.
Discussion:
"Developing biomarkers that can identify NMIBC patients likely to progress for early radical cystectomy or neoadjuvant/adjuvant chemotherapy to improve outcomes"
something seems to be wrong with that sentence. What do you mean? Or is "that" too much...?
"Based on a search in PubMed, no previous study has investigated the effects on gene
expression of exposure to 1%, 0.2% and 0.1% O2 concentrations levels in multiple bladder
cancer cell lines" --> typo??
"A possible criticism is that the study only examined cells exposed to hypoxia for a single time, 24h"
I assume it to be very positiv that this aspect it discussed. Instead of only discussing that it "represents biologically relevant hypoxia" it would be helpful to discuss literature, where 24h hour hypoxia was compared to multiple cycles of hypoxia and to extrapolate the differences and limitattions these data means for your work. See literature below.
Olbryt M, Habryka A, Student S, JarzÄ…b M, Tyszkiewicz T, Lisowska KM. Global gene expression profiling in three tumor cell lines subjected to experimental cycling and chronic hypoxia. PLoS One. 2014 Aug 14;9(8):e105104. doi: 10.1371/journal.pone.0105104. PMID: 25122487; PMCID: PMC4133353.
Please discuss how the data can be incorporated into clinical practice in the future. What are the plans for the future? whats the next step? Animal models, etc.?
Author Response
Dear Reviewer,
Thankyou for you helpful comments which I have attended to as follows:
Reviewer 2:
- I have changed the second sentence to not be specific regarding proportion of patients with MIBC treated with RT as I’m finding it difficult to come up with a figure. I have also replaced reference 3 with the one you suggested. It is much clearer the proportion of patients who receive RC.
- Typos and poor grammar:
Discussion
- ‘Developing biomarkers that can identify NMIBC patients likely to progress for early radical cystectomy or neoadjuvant/adjuvant chemotherapy to improve outcomes’ something seems to be wrong with that sentence. What do you mean? Or is "that" too much...?
- "Based on a search in PubMed, no previous study has investigated the effects on gene
expression of exposure to 1%, 0.2% and 0.1% O2 concentrations levels in multiple bladder cancer cell lines" --> typo??
- "A possible criticism is that the study only examined cells exposed to hypoxia for a single time, 24h"
We have now corrected these sentences.
- I assume it to be very positive that this aspect it discussed. Instead of only discussing that it "represents biologically relevant hypoxia" it would be helpful to discuss literature, where 24h hour hypoxia was compared to multiple cycles of hypoxia and to extrapolate the differences and limitattions these data means for your work. See literature below.
Now included and discussed this useful paper: Olbryt M, Habryka A, Student S, JarzÄ…b M, Tyszkiewicz T, Lisowska KM. Global gene expression profiling in three tumor cell lines subjected to experimental cycling and chronic hypoxia. PLoS One. 2014 Aug 14;9(8):e105104. doi: 10.1371/journal.pone.0105104. PMID: 25122487; PMCID: PMC4133353.
- Please discuss how the data can be incorporated into clinical practice in the future. What are the plans for the future? whats the next step? Animal models, etc.?
We have added a short paragraph before the final summery paragraph in the Discussion regarding the use of a further NMIBC cell line and analysis of gene expression of in vivo models of the cell lines for future work.
Reviewer 3 Report
Comments and Suggestions for Authors
Revisions required for publication
- In regard to the method section 2.3, the authors report that cells were given a 24h recovery and controls were incubated for 24h in normoxic conditions, but the authors write “(acute hypoxia)”, this needs to be revised to specify the conditions used.
- Authors describe hypoxic cells being washed two times 24h prior the incubation, since there was a 24h recovery time after trypsin, were there 24h after the recovery time and since cells were incubated for 24h in hypoxic conditions, were these cells maintained for longer than controls?
- In section 2.5 authors describe two extraction methods, which were they used in all conditions? Were RNA extraction methods compared? If RNA extraction methods were used for all conditions or specific conditions how did this affect the results from ClariomTM S microarrays?
- In section 2.6 and 2.7 authors describe that differential analyses were performed with DESeq2 and limma_3.48.3. Since different methods were described, were the results different? Which results were provided for the result section?
- Section 2.11 describes the datasets used. Was there consideration of NMIBC and MIBC? A table of clinic-pathologic data should be provided since the authors aim to identify differences in NMIBC and MIBC, and specifically, aiming for biomarkers for early detection, datasets must comprise these characteristics.
- Authors chose 4 MIBC and 2 NMIBC cell lines, more NMIBC should be considered for clustering purposes and better gene panel identification as well as primary cell line to identify disease specific genes.
- Authors throughout the manuscript describe genes found in at least three cell lines; were these cell lines only MIBC? Since the intent is to detect different patients and to exploit the differences between NMIBC and MIBC, this choice doesn’t seem to be beneficial for this purpose
- The manuscript would benefit from identifying DEGs for NMIBC and MIBC individually, for every studied condition, and comparing them to the already described 24-gene panel, as well as identifying common genes between groups.
Optional and minor work recommended to authors
Table 1, authors should display to which cell lines the number of genes represents
- Biological processes could be presented as images, such as the ones produced by Cytoscape for better interpretability, and tables could be supplied as supporting information
Authors should show volcano plots based on DEG analysis for better visualisation and sample grouping understanding
Comments on the Quality of English LanguageThere should be an English revision concerning the scientific soundness of the manuscript, for example, in lines 70-71 "No one has investigated how use of different oxygen levels might affect candidate gene panels and ability to prognosticate " should be changed to "To our knowledge, there haven't been described how different oxygen levels might affect the identification of gene panels and their prognostic value"
Author Response
Dear Reviewer
Thank you for your suggestions which have enhanced the clarity of the manuscript. We have addressed your suggestions as follows:
- In regard to the method section 2.3, the authors report that cells were given a 24h recovery and controls were incubated for 24h in normoxic conditions, but the authors write “(acute hypoxia)”, this needs to be revised to specify the conditions used.
Thank you this was confusing and has now been clarified. The procedure involved incubation of cells in normoxia for 24h post-plating, the media was then either replaced with fresh media and the cells incubated in normoxic conditions or replaced with ‘hypoxic’ media in the hypoxia cabinets and incubated under hypoxic conditions - in either case for a further 24h.
- Authors describe hypoxic cells being washed two times 24h prior the incubation, since there was a 24h recovery time after trypsin, were there 24h after the recovery time and since cells were incubated for 24h in hypoxic conditions, were these cells maintained for longer than controls?
Please see response to comment 1. For both normoxia and hypoxic treatments an initial 24h period in normoxia was carried out then either media was replaced with fresh media in normoxia or hypoxic medium in hypoxic conditions for 24h.
- In section 2.5 authors describe two extraction methods, which were they used in all conditions? Were RNA extraction methods compared? If RNA extraction methods were used for all conditions or specific conditions how did this affect the results from ClariomTM S microarrays?
Both procedures are used for each purification – however the QIA shredder is used first to disrupt the cells then the RNAeasy. The order is now correct and the reasons for the use of each is described.
- In section 2.6 and 2.7 authors describe that differential analyses were performed with DESeq2 and limma_3.48.3. Since different methods were described, were the results different? Which results were provided for the result section?
Apologies we only used limma-3.48.3 for this study. Reference to DESeq2 removed.
- Section 2.11 describes the datasets used. Was there consideration of NMIBC and MIBC? A table of clinic-pathologic data should be provided since the authors aim to identify differences in NMIBC and MIBC, and specifically, aiming for biomarkers for early detection, datasets must comprise these characteristics.
The characteristics of the 6 cell lines are now summarised in table 1.
- Authors chose 4 MIBC and 2 NMIBC cell lines, more NMIBC should be considered for clustering purposes and better gene panel identification as well as primary cell line to identify disease specific genes.
It would be interesting to have more NMIBC cell lines. RT4 followed by RT112 used in this study are the two commonly used NMIBC lines. There is only one other NMIBC cell line available (SW780). One paper that has looked at both primary and established cell lines highlighted 8 genes upregulated in >5 of 32 primary bladder cancer tissue that are upregulated by hypoxia in vitro. In our study we found that 7 of these genes were upregulated in at least 3 of the bladder cancer lines or in in one case at least two of the MIBC cell lines but not in NMIBC cells in our study. We have added this to the discussion (penultimate paragraph)
- Authors throughout the manuscript describe genes found in at least three cell lines; were these cell lines only MIBC? Since the intent is to detect different patients and to exploit the differences between NMIBC and MIBC, this choice doesn’t seem to be beneficial for this purpose
We used genes upregulated in at least 3 cell lines (MIBC and NMIBC) as a basis for identifying possible candidate genes for hypoxia biomarkers. However for the analysis of differential up-regulation of genes between MIBC and NMIBC we identified genes expressed in both NMIBC cell lines and compared these with genes upregulated in at least two MIBC cell lines. This is now described in Methods 2.7.
The manuscript would benefit from identifying DEGs for NMIBC and MIBC individually, for every studied condition, and comparing them to the already described 24-gene panel, as well as identifying common genes between groups.
This has now been carried out and the analysis summarised in supplementary table 1. This was an interesting analysis demonstrating core genes upregulated in all cell lines, some just in one or two and two genes not in any. This has been added to the results section.
Optional and minor work recommended to authors
Table 1, authors should display to which cell lines the number of genes represents
Table 2 (was table 1) includes the names of the cell lines used in the legend.
Biological processes could be presented as images, such as the ones produced by Cytoscape for better interpretability, and tables could be supplied as supporting information. Authors should show volcano plots based on DEG analysis for better visualisation and sample grouping understanding
We have not had time to implement these changes but appreciate these suggestions.
Reviewer 4 Report
Comments and Suggestions for Authors
This study "Gene expression in muscle invasive and non-muscle invasive bladder cancer cells exposed to hypoxia" employs transcriptomic profiling of bladder cancer cell lines exposed to varying hypoxic conditions, aiming to identify hypoxia-responsive genes with potential prognostic relevance. While the research topic is highly relevant and the in vitro model is well-justified, the manuscript suffers from several significant methodological and interpretative limitations that must be addressed before publication.
1.The authors describe RNA quantification via Nanodrop and Qubit, but do not provide RNA integrity metrics, such as RIN values obtained from a Bioanalyzer or TapeStation. Even in cell culture-based transcriptomics, RNA quality must be demonstrated to ensure that gene expression profiles reflect biological rather than technical variation.
2.The number of biological and technical replicates per condition or per cell line is not reported. Replication is critical in transcriptomic studies—even those based on cell lines—to assess variability and ensure statistical robustness. This omission limits the ability to interpret the strength and reproducibility of the findings.
3.While the manuscript mentions the inclusion of brain, prostate, and bladder cancer reference RNAs, their precise function in the experiment remains unclear—whether they were used for normalization, batch correction, or as external quality controls. This should be explicitly described to ensure interpretability.
4. The description of normalization using “single space transformation with probe guidance cytosine count correction” is unusual and lacks standard citation or explanation.
5. The authors provide no information on microarray QC metrics such as background signal distribution, internal control probe performance, or signal-to-noise ratios. Reporting such metrics is essential for evaluating data quality in microarray experiments.
6.The findings, including identification of novel hypoxia-associated genes, are not supported by any independent biological validation such as qPCR or protein-level confirmation. While such validation may be outside the scope of an exploratory in vitro study, some confirmation of key DEGs would strengthen the conclusions.
7. Although statistical packages like limma and DESeq2 are used, the workflow lacks a clear and logical presentation. The justification for using different significance thresholds (padj < 0.001 for biomarker selection and < 0.05 for subgroup comparisons) is not discussed.
8. The authors declare that data will be deposited in GEO, but do not provide a specific accession number, persistent link, or timeline. This does not meet FAIR data principles. For the manuscript to be considered fully transparent and reproducible, data must be made publicly available at the time of review, or a GEO submission ID must be included.
Author Response
Dear reviewer
Thank you for your comments. We have addressed these as follows
- We use 260/280 and 230/260 absorbance ratios as a measure of RNA quality. The mean and range of these values across the samples is now included in section 2.4.
- For each cell line/condition we carried out the experiment 3 times using different cell passages yielding 3 biological replicates. This is now stated more clearly in section 2.3
- The cell samples were run with clinical samples which were part of a series and the other tissue controls were included to ensure there reproducibility across batches and were irrelevant for this study so have been removed.
- The process described was recommended by the manufacturer Thermofisher. We have now included a link to their white paper: https://assets.thermofisher.com/TFS-Assets/LSG/brochures/sst_gccn_whitepaper.pdf
- Microarray was outsourced to Yourgene, Manchester. They carried out QC on each sample. We received only a QC pass or fail. Samples used in this study had all passed their QC criterion.
- Although we did not carry out PCR as part of this study, we have previously shown that the 24 gene hypoxia signature is sensitive to hypoxia (0.1% O2) in RT112 and J82 cells.
- Apologies we only use limma for this study and have now removed DESeq2 from the methods section. The justification for using two levels of significance is now included in methods 2.7
- The transcriptome is now deposited in the GEO repository - Accession number GSE302555 (added to data availability statement)
Round 2
Reviewer 3 Report
Comments and Suggestions for Authors
The authors have answered all major suggestions to the manuscript
Comments on the Quality of English LanguageI find that there should be a revision of the English concerning scientific soundedness